# An Exploratory Typology for Understanding Family-Relationship Issues in Kinship-Care Placements

**Amilie Dorval [1,\*], Sonia Hélie [2] and Marie-Andrée Poirier [3]**

1. Department of Psychoeducation and Social Work, Université du Québec à Trois-Rivières, Drummondville, QC J2C 0R5, Canada

2. Institut Universitaire Jeunes en Difficulté, Montréal, QC H2L 4R5, Canada; sonia.helie.ccsmtl@ssss.gouv.qc.ca

3. School of Social Work, Université de Montréal, Montréal, QC H3C 3J7, Canada; marie-andree.poirier@umontreal.ca

\* Correspondence: amilie.dorval@uqtr.ca

**Abstract:** Family relationships are a distinctive feature of kinship-care placements, but very few studies have examined how the dynamics of these relationships affect the placement experience. This article does explore these dynamics and identifies some possible patterns, as experienced and reported by parents of children placed in kinship care. The findings presented here come from a qualitative study employing a life-story methodology, in which nine parents were interviewed on two occasions each. All of them had experienced the permanent placement of at least one of their children with a member of their extended family, under the direction of a government child-protection agency. Drawing from significant themes in parental narratives, particularly that of relationships, we analyzed and delineated three distinct profiles. In the first profile, a family solidarity was present between the parents and the kinship caregivers before the placement and was maintained during the placement. In the second, the parents struggled to keep their place in their children's lives, thus experiencing conflicts both with the kinship caregivers and with the child-protection agency. In the third profile, the dynamics of the current relationship between both biological parents influenced the other family relationships of the parent who was interviewed.

**Keywords:** kinship care; parents; relationship; youth protection

## 1. Introduction

In the Canadian province of Quebec, as in many other jurisdictions, when children have to be removed from their family settings, the preferred option is to place them in kinship care, which means care in a setting under the responsibility of someone who has a significant connection with but may or may not be related to the child [1,2]. This practice supports the importance of family bonds and their maintenance during placement. Several studies have reported an increase in kinship-care placements in many countries since the early 2000s [3–6].

There are two types of kinship-care placements: formal placements, which are planned and managed by the child-protection authorities, and informal placements, which are arranged within the family without these authorities' being involved. In Quebec, very few records are kept regarding informal kinship-care placements, so most of the studies on kinship-care placements have been based on formal placements alone. Consequently, the data presented in this article relate to formal kinship care placement.

Some studies have addressed the distinctive aspects of family relationships in kinship-care placements and the issues that they involve [3,7–12], but very few studies have examined the relationships between the biological parents and the kinship caregivers, as experienced and reported by the biological parents. It seems important to document the experience of these parents, who remain central figures in their children's lives even after the placement. The findings presented in this article come from a study that more

broadly documented the experience of parents whose children had been permanently placed with members of their extended families by Quebec's child-protection authorities. More specifically, this article presents findings regarding the relationship aspects of such placements.

## 2. Review of the Literature

One of the main distinctive features of kinship-care placements is the connection that already exists not only between the kinship caregiver and the child, but also, by extension, between the kinship caregiver and the biological parent. The literature shows that in many situations, the kinship caregiver is part of the parent's family—either an immediate family member (such as a mother, father, brother, or sister) or an in-law [1,3,4,13]. The relationship between the biological parent and the kinship caregiver may change in various ways following the placement decision. Studies that have looked at this relationship have identified various dynamics that may come into play and influence the experience of parenthood during a kinship-care placement.

One possible dynamic is that communication may be easier with a kinship caregiver: in a study in Australia, some biological parents stated that they preferred for their children to be placed with kinship caregivers rather than with regular foster caregivers, in particular because of the connections that their children had with these people, the possibility of maintaining a connection within the family, and the support received by the kinship caregiver [7].

A study of biological parents who experience challenges with drugs and alcohol, in Scotland, showed that the relationship between such parents and kinship caregivers may change during the placement, especially if the kinship caregiver is one of the parent's own parents [14]. The child's biological parents in this study report being grateful for the help and support they received from their families, who took charge of their child during moments of great instability due to consumer issues, such as housing instability. They added, however, that the kinship caregiver often behaved in ways that were perceived as surveillance towards them, and some felt "infantilized" and impaired in their parenting skills. These relational issues were particularly reported by biological parents whose kinship caregiver was their own parents. The results of this article highlight several relational difficulties experienced between the parents and the kinship caregiver, but also possible changes in the relationship. For instance, some relationships may become highly conflictual, while others may cease to exist altogether.

In another study conducted in 2011 [15], in which children whose biological mothers were incarcerated had been placed with their maternal grandmothers, the authors interviewed the mothers and grandmothers concerned, and two opposing patterns clearly emerged: in some cases, the two women felt a strong sense of solidarity in their co-parenting relationship, while in others, they felt a lack of solidarity. Obviously, solidarity is not an all-or-nothing phenomenon; it must be measured along a continuum. At one end of the continuum, the mothers and daughters saw their co-parenting relationship as very helpful and characterized by co-operation, good communication, mutual support, a search for shared solutions, and a similar philosophy of child-rearing. But at the other end, the mother–daughter relationship may have undermined the placement. Conflict in this relationship (in particular, fights in front of the child, disagreements, and constant criticism of each other) sometimes created problems in taking care of the child. Other examples of lack of solidarity between mothers and grandmothers with whom children had been placed included diverging views about discipline, grandmothers' failure to meet their commitments, and mothers' disengagement from parenting.

Similarly, the problem most often reported by the biological parents interviewed in 2015 by Kiraly and Humphreys [7] was their complex, ambivalent relationship with their children's kinship caregivers (in 50% of these cases, this meant the relationship between the children's mother and her own mother). A few of these biological mothers expressed resentment toward their own mothers, believing that their own experience in childhood had

contributed to their current difficulties in parenting. Having their children placed in their mothers' care would appear to have aggravated their resentment and anger toward them.

Ross et al. [16] showed that relations between out-of-home caregivers and biological parents are often more difficult in kinship-care placements than in regular foster placements. These authors also underscored that child-protection authorities do not have a great deal of control over such relationship issues, because many contacts may occur informally.

The literature has also shown that the quality of the relationship between the biological parent and the kinship caregiver affects the parent's commitment and involvement during the child's placement. In Ross et al. [16], some biological parents said that they considered it important to feel respected and recognized by the out-of-home caregivers, and even that their commitment and involvement with their children depended greatly on their own relationship with these caregivers. Some biological parents also said that their children's caregivers (whether foster parents or kinship caregivers was not specified) had even surpassed the child-protection agencies' requirements in making a place for them in their children's lives. Similarly, Poirier and Simard [17] found that a positive perception of the biological parents by the kinship caregivers was associated with greater parental involvement.

Holtan and Eriksen [18] showed that the kinship caregivers often let the biological mother play a greater role, in particular because the connection that the parent, the child, and the kinship caregiver already had before the placement continued during it. In contrast, these authors found that some regular foster families showed openness toward the biological parents and gave them a greater role at the start of the placement, but tended to show less openness over time, whereas kinship caregivers tended to remain more open to the parents.

Differential approaches to parental involvement emerge between kinship caregivers and regular foster families. While kinship caregivers tend to sustain openness towards biological parents throughout the placement, regular foster families may exhibit waning openness over time [18]. This enduring openness in kinship-care placements underscores the continuity of pre-existing connections, fostering a collaborative environment conducive to shared parenting roles.

Lastly, on the basis of a study that she performed in the Republic of Ireland decades ago, O'Brien [8] presents a model of the relationship dynamics in kinship care placements. In this model, O'Brien [8] identifies four main sets of actors (the biological parents, the caregivers, the child, and the child-welfare agency) and two major relationship patterns (co-operative and conflictual). She divides the co-operative pattern into the sub-categories "shared care" and "quasi-adoption", and the conflictual pattern into the sub-categories "oscillating" and "distressed".

In shared care, as O'Brien [8] describes it, parenting is shared between the caregivers and the biological parents, all of whom co-operate to serve the best interests of the child. In this sub-category of relationship patterns, the child-welfare agency stays at the periphery and intervenes very little. The factors associated with shared care include older children, voluntary placement status, a history of informal care within the family, absence of conflict over access to the child, and the agency's being satisfied that the child's protection needs are adequately safeguarded.

In the quasi-adoption pattern, as the name suggests, the biological parents are absent. The child-welfare agency collaborates with the caregivers to ensure the child's safety and development. The reasons that the child-welfare agency remains involved and that the child is not placed for adoption may include the caregivers' financial needs and the services that the child requires. O'Brien [8] notes that the quasi-adoption pattern may be seen when an informal placement in kinship care is moved into the formal care system or in some cases when the kinship-care placement proves stable over the long term and the biological parents tend to disappear from the equation.

In the oscillating pattern, the relationship dynamic between the biological parents and the caregivers alternates between cooperation and conflict. The parents ally themselves

with the caregivers at some times and with the agency at others. O'Brien [8] reports that this pattern is often seen when there is no explicit long-term care plan for the child: the parents' ambivalence about the permanence of the placement and their hope of getting their child back may cause their position to fluctuate.

In the distressed relationship pattern, the parents tend to be excluded, but in this case without their consent. Thus, in contrast to the quasi-adoption pattern, in which the parents are absent more by their own choice, the parents involved in a distressed dynamic do not accept their exclusion and come into conflict with both the agency and the caregivers. O'Brien [8] states that a distressed relationship dynamic is not usually seen at the start of a kinship-care placement, because the agency would not make such a placement under such circumstances. Instead, this dynamic results from conflicts that develop after the placement has been made.

In summary, the reviewed literature demonstrates that the intricacies of the relationship between biological parents and kinship caregivers extend beyond mere interaction, significantly impacting the parent's commitment and involvement during the placement. Studies emphasize the pivotal role of respect and recognition accorded by out-of-home caregivers in shaping parental engagement. Notably, the quality of this relationship often dictates the extent of parental involvement, transcending prescribed agency mandates [16,17].

Despite the evident influence of the relationship between biological parents and kinship caregivers on the placement experience, scant research beyond O'Brien's typology has explored this crucial aspect comprehensively. Notably absent are studies elucidating, from the biological parents' standpoint, the intricate interplay of relationship dynamics and their impact on the placement experience. Hence, the purpose of this article is to present a typology of these relationship dynamics according to the experiences reported by the biological parents of children who have been permanently placed with members of their extended families.

## 3. Methodology

Working with a provincial-government child-protection agency in the Greater Montreal metropolitan area, we recruited nine biological parents, each of whom had had at least one child placed until the age of majority with a kinship caregiver who was one of the child's direct relatives (an uncle, aunt, or grandparent). As mentioned above, the placements are all formal kinship care. The agency's clinical teams made first contact with these parents to recruit them as agreed by the research ethics board[1] and obtain their consent to be interviewed for this study. (Some of these parents had been previously identified from a placement cohort in another study that was in progress at the time of recruitment [12]). Once the parents had agreed to be contacted by the research team, the author contacted them and made an appointment with each of them at a location of their choice.

Using a life-story methodology, we then conducted two qualitative interviews with each of these nine parents (except for one mother whom we interviewed only once). In the first interview, the researcher obtained and constructed the parent's life story. In the second interview, the researcher presented her understanding of the parent's life story to the parent and obtained further information on some items that had come up in the first interview. The interview guide provided only a few questions, so that the interviewer could let the parents elaborate on the things that they considered meaningful. The main themes covered were guided by the theoretical framework for this study—the theoretical model of parenthood presented by Houzel [19] and Sellenet [20]. According to this model, parenthood is based on three dimensions: experience, practice, and exercise. Parenthood experience encompasses the psychological aspects, including the parent's connection to the child, ideas about parenthood, emotions and cognitions, as well as the parent's own assessment of his or her parenting. Parenthood practice consists of the acts of care that the parent must provide and that comprise daily life, shared moments, and direct and indirect contact with the child. Lastly, parenthood exercise is based on the legal aspects of parenthood, including parental authority and responsibility. The themes in the interview

guide included the parent's life story before the placement; relationship with the kinship caregiver (and how it evolved before and during the placement); parenthood experience, parenthood practice and parenthood exercise during the placement; experience with the child-protection agency; and outlook for the future. The interviews ranged in length from 76 min to 3 h (mean = 131 min), and the time between interviews ranged from 4 to 16 weeks (mean = 7.86 weeks). All of the interviews were transcribed. Table 1 summarizes the characteristics of the nine parents interviewed.

**Table 1.** Summary data on the parents interviewed.

| Name | Age at Time of Interview | Marital Status | Employment Status | Number of Children | Kinship Caregiver(s) |
|---|---|---|---|---|---|
| Martha | 28 | Single | Unemployed | 1 | Maternal grandmother |
| Stephen | 30 | In a relationship | Unemployed | 1 | Maternal grandmother |
| Virginia | 28 | In a relationship | Employed | 2 | Maternal grandmother Maternal grandmother |
| Joanie | 38 | In a relationship | Employed as of second interview | 2 | Maternal grandmother Father's cousin (godmother) |
| Nicholas | 35 | In a relationship with the mother of his second child | Unemployed | 2 | Mother Paternal aunt |
| Samantha | 31 | In a relationship with the father of her second child | Unemployed | 2 | Maternal grandfather (deceased) and his spouse Mother |
| Amelia | 28 | Uncertain | Had become employed recently | 2 | Paternal grandmother Mother |
| Carol | 53 | In a relationship | Unemployed | 1 | Paternal aunt |
| Jason | 34 | In a relationship | Employed | 2 | Maternal aunt Mother's female cousin |

*Analyses*

Using NVivo software version 11, we subjected all nine parents' interview transcripts to a classic form of thematic analysis (see, for example, [21]). As described by Paillé and Mucchielli [22], a thematic analysis consists in systematically identifying the themes addressed in a body of discourse, classifying them into groups, and then analyzing the discourse in light of these themes. In the present study, the thematic analysis identified several relevant themes, which we then classified into groups according to Houzel and Sellenet's theoretical model of parenthood. A validation of the coding tree was carried out between the first author and a graduate student, as well as a consensus-based coding with two interviews.

In addition to analyzing and grouping the themes according to these three dimensions of parenthood, we analyzed and grouped the factors that might have influenced the parents' experience. Four major categories of themes thus emerged from our analysis of the parents' stories. These themes are shown in Table 2; the themes that differentiated the three groups of parents described below are identified by bold characters and an asterisk in this table.

To gain a deeper understanding of the parents' experiences beyond simply describing the themes identified in their discourse, we performed another analysis whereby we defined a typology of these experiences (see Table 3). The typology revealed three distinct groups: 1. Family solidarity; 2. Parental struggles; 3. Conjugal–Parental Dyad. This typology shed additional light on the relationship dynamics by grouping together those parents who seemed to have had similar experiences, on the basis of a cross-analysis of their stories. This exercise gave us an overview of the material compiled and let us identify similar patterns in the parents' responses, using a table with one column for each parent and one row for each of the various themes that had emerged from the analysis.

Upon inspection, this table clearly revealed two groups of parents (the first and second group, respectively family solidarity and parental struggles), with a sharp contrast between these groups but a degree of consistency within them. Once we had identified these two groups, we conducted a deeper analysis which showed that the experience of one of the parents diverged from those of the parents in the two groups, so we had to create a third group. The themes that we used to classify the parents into groups dealt both with the parents' relationships with the other persons concerned (a factor specific to kinship-care placements that proved to be very central and highly distinctive) and with the parenthood dimensions of the theoretical model that guided this research.

**Table 2.** Themes identified in the thematic analysis, according to Houzel's dimensions of parenthood.

| Experience Dimension | Practice Dimension | Exercise Dimension | Influencing Factors |
|---|---|---|---|
| 1. Desire to have children and experience parenthood<br>2. Removal experience<br>3. Reactions to placement and its consequences<br>4. Cognitions expressed<br>5. Failed reunification experience<br>6. Experience of placement until age of majority<br>7. Partial-parenthood experience<br>8. Projections into the future | 1. **Problems experienced before the placement ***<br>2. **Experience of cohabitation with the kinship caregiver ***<br>3. **Contacts with the child ***<br>4. Communication with the child regarding the placement<br>5. Other forms of involvement<br>6. Positive elements of parenthood practice | 1. Decision-making<br>2. **Concepts of parental rights, responsibilities, and authority ***<br>3. **Interactions with the justice system ***<br>4. **Lack of recognition ***<br>5. **Parent's sense of filiation with child *** | 1. Biographical events<br>2. **Relationships ***<br>　A. **With the kinship caregiver**<br>　B. **With the child-protection agency**<br>　C. **With the conjugal partner** |

Note: * Theme that differentiated the groups.

Table 3. Summaries of the parents' discourses on the various themes that differentiated them into the three groups shown.

| | | Family Solidarity | | | | Parental Struggles | | | | Conjugal–Parental Dyad |
|---|---|---|---|---|---|---|---|---|---|---|
| | | **Martha** | **Virginia** | **Joanie** | **Samantha** | **Stephen** | **Carol** | **Amelia** | **Jason** | **Nicholas** |
| Relationships | With kinship caregiver | Fairly positive | Fairly positive | Positive | Positive Improved | Very difficult | Very difficult | Ambivalent | Difficult | Ambivalent and deteriorated |
| | With child protect-tion agency | Little to say | Little to say | Talked about a certain child-protection worker | Little to say except about staff turnover | Difficult: talked a lot about the agency as a threat and posing obstacles | Difficult: says she did not receive enough support, did not receive services to help with her substance use | Difficult: talked a lot about the agency as a threat, abusive, and posing obstacles | Difficult: says he was not recognized as a father, and that some child-protection workers were incompetent | Difficult: plan for his son's adoption or tutorship, which he did not accept |
| | With other parent | No recognized father | Major conflict. Separated | Major conflict. Separated | Little to say. Separated. | Major conflict. Separated. | Little to say. Separated. | Little to say. Separated. | Major conflict. Separated. | Conflictual, but still in a relationship |
| Themes under Dimension 2, Parenthood Practice | Problems with the place-ment | Admitted her problems before the placement | Admitted her problems to some extent | Admitted that she had some problems before the placement and that her son presents some major challenges | Admitted her responsibility in the placement and her substance-use problem | Somewhat admitted her problems in taking care of her newborn son | Somewhat admitted her substance-use and other problems | Did not admit her problems, denied them | Admitted having engaged in criminal behaviour but denied its impact on his children | Admitted his problems to some extent, said that he had an intellectual deficit and anger-management problems |
| | Experience of cohabitation with caregiver | Cohabitation at start of placement | No cohabitation | Frequent cohabitation | Cohabitation at start of placement | No cohabitation | No cohabitation | No cohabitation | No cohabitation | Cohabitation at start of placement |
| | Contacts with children | Contacts not supervised Overnight stays allowed | Contacts supervised by kinship caregiver Contacts increased | Contacts supervised by kinship caregiver | Contacts not supervised | Contacts supervised by child protection Frequency decreased | Contacts not supervised Overnight stays allowed | Contacts not supervised | Contacts supervised by child protection Contact stopped following his incarceration | Contacts supervised by child protection Regular intervals |
| Themes under Dimension 3, Parenthood Exercise | Concepts of parental rights, responsibilities and authority | Signed important documents | Nothing to say | Confusion between physical and legal custody | Nothing to say | Nothing to say | Said she had legal custody. Signed papers Right to assistance services | Often referred to her rights, right to have contact | Nothing to say | Referred to his rights a few times |
| | Interactions with the justice system | Nothing to say | Nothing to say | Difficult experience before going to court | Nothing to say | Nothing to say | Addressed. Experience difficult, did not understand the system or the vocabulary | Addressed. Experience with judge and lawyers | Nothing to say | Developed. Difficult experience of his efforts' not being recognized |
| | Lack of recognition | Little to say, except in the making of certain decisions | Nothing to say | Nothing to say | Nothing to say | Lack of recognition as a father | Lack of recognition of her status as a mother | Lack of recognition of her dignity and humanity | Lack of recognition of the efforts he had made | Lack of recognition of the efforts he had made, said he was not listened to or considered |
| | Filiation | Not very present | Not very present | Not very present | Not very present | Very present; biological filiation | Very present; references to physical resemblance | Present; references to physical resemblance | Very present; reference to physical resemblance | Very present; sense of biological filiation broken by adoption plan |

## 4. Results

To help provide the clearest possible picture of our analyses of the nine parents' life stories and the three groups that we derived from them, Table 3 summarizes, by theme, the things that these parents told us that let us classify them into these groups. This table provides a highly condensed analysis of these stories and so does not convey all of the details and subtleties. But it does give a good idea of what the stories of the parents in each of the three groups have in common.

### 4.1. Group 1: Family Solidarity

The parents in Group 1, Family Solidarity, had fairly positive relations with their children's kinship caregivers and with the child-protection agency. This group consisted of four mothers: Martha, Virginia, Joanie, and Samantha. Each of them had one or two children, all of whom had been placed on the mother's side of the family, except for Joanie's second son, who had been placed with a female cousin on his father's side. Martha's, Virginia's, and Joanie's children had been placed with their maternal grandmothers, whereas Samantha's had been placed with their maternal grandfather and, following his death, with his spouse.

The two distinctive characteristics of the parents in this group are that they had good relations with their children's kinship caregivers and that the child-protection agency played little role in their stories. Each of these four mothers described a fairly harmonious relationship with the kinship caregiver. All of them said that they received support from her and saw her frequently. The following quote from Martha's interview is a good example of this supportive relationship with the kinship caregiver:

> Yes, [my mother and I talk], she helps me a lot, because I have very limited mobility. For example, on Monday, she gave me a lift to my doctor's, because she knew that if she didn't take me, I wouldn't go. On Sunday, she also took me to the neurologist for my magnetic resonance. She helps me a lot.

Except for Samantha, these mothers felt that the placement had not altered their relationship with the caregiver—that it had been good before and continued to be good afterward. Samantha, on the other hand, said that she had not really developed any relationship with her father's spouse before her children were placed with her, and had not even trusted her particularly. But Samantha acknowledged having developed a relationship with her father's spouse since this time, whom she considers as her stepmother. She explained that her stepmother let her continue to play a considerable role in her son's life and that the relationship between these two women had grown stronger since the placement: now Samantha sees her stepmother and receives help from her often, and they communicate regularly. As she relates:

> I trusted my father when he was taking care of my son. I even told him, "You're the only one I trust." But then he died, and after that, I began to trust my stepmother. She helped me a lot then, she helped me with my housing and all that. Especially when she told me that she would take care of my son for me anyway. [...] [Now] things are going well, we talk to each other. She's always there to help us, if there's anything we need, [she helps us] if she can. [We communicate well about my son.]

The second distinctive characteristic of the four mothers in the Family Solidarity group is that they mentioned the child-protection agency very seldom in their stories. They had little to say about their relations with this agency and very little criticism to offer about it. They also had very little to say about their child-protection workers, other than the same things that the parents in the two other groups mentioned: that the agency's staff turnover was very high, and that they liked some of these workers more than others. These four mothers' stories also indicated that this agency was not very involved in supervising their contacts with their children. For example, Virginia and Joanie said that they had some contacts with their children that were supervised by the kinship caregiver and not

by a child-protection worker, while the two other mothers in this group said that they had unsupervised contacts with their children, who were allowed to stay overnight with them.

Regarding their relations with their children's fathers, all four of these mothers said that these fathers were not very present or involved. But although most of these mothers had negative things to say about their children's fathers, this relationship did not seem to be affecting them particularly at the time of the interviews.

The contacts between the mothers in this group and their children seem to be freer and less formalized by the child-protection agency. For three of these mothers, the frequency of these contacts seemed fairly stable; however, in Virginia's case, it had been increasing in the months preceding the interviews. Virginia explained in her story that in the past, she had completely cut off any contacts with her children and her family, but that she had now resumed such contacts. Also, as mentioned previously, these contacts were supervised for two of the mothers (Joanie and Virginia) but not for the two others (Martha and Samantha). However, in contrast with the other groups, in this group the person responsible for supervising the visits was the kinship caregiver, and the mothers in question reported that this caregiver took a back seat during their contacts with their children.

Lastly, the mothers in this group had very little to say on the themes related to the third dimension of parenthood examined in our analysis: parenthood exercise. In contrast, as we shall see in the following section, these themes loomed large in the interviews with the parents in Group 2.

*4.2. Group 2: Parental Struggles*

Unlike the parents in Group 1, the parents in Group 2, Parental Struggles, had fairly negative relationships both with their children's kinship caregivers and with the child-protection agency. Like Group 1, Group 2 comprised four parents: Jason, Stephen, Amelia, and Carol. Each of these parents had either one or two children, all of whom had been placed, except for Amelia, who was experiencing partial parenthood at the time of the interviews: her older daughter had been placed, but she still had custody of her second child. All four of the parents in this group had had their children placed with their former in-laws' side of the family, and all four said that in their role as parents, they were not receiving any support from either the kinship caregiver or the child-protection agency. In sharp contrast with the parents in the Family Solidarity group, those in the Parental Struggles group described tense relationships with their children's kinship caregivers. Three of these parents talked about having a difficult relationship with the kinship caregiver, generally characterized by mutual disdain, as evidenced by the following quotes:

> Because, first of all, well, I wouldn't say we hate each other, but we don't like each other. (Stephen)

> My relationship with the aunt was tough, because she would have loved it if I wasn't on the scene at all and she didn't have to be involved with me, [having to bring my daughter to visit me and experience all that]. She thought that she'd have her peaceful little life with my daughter [without having to have me around]. (Carol)

Another of the parents in Group 2, Amelia, provided a more nuanced picture of the paternal grandmother who was acting as kinship caregiver for her daughter. She expressed mixed opinions about her, mentioning some events that had shaken her confidence, as well as various conflicts, notably in relation to diverging views on how to raise her daughter. Amelia's story, which does not really describe a supportive, positive overall relationship with the kinship caregiver, thus differs from the stories of the mothers in Group 1.

The second thing that distinguished the parents in Group 2 from those in Group 1 was definitely their difficult relationship with the child-protection agency. In fact, their relationship with this agency and its workers was central to their stories. Three of the four parents in Group 2 spent a substantial part of their interviews talking about their child-protection workers and their negative experience with the child-protection system.

All four of these parents said that they felt that the system did not hear them or take them into consideration and seemed to be constantly putting obstacles in their way.

> I didn't agree [with the placement to age of majority], but at the same time, I didn't have a choice. Because that's [what was going to happen], whether I agreed or not. [What I learned from that is] if you want to say something, say it, and if you don't, don't, it doesn't matter, it's going to happen anyway. (Stephen)

> But they didn't give a shit. They didn't want to hear anything about my evidence, that didn't matter to them at all. (Amelia)

The parents in this group seem to be driven by a certain anger and bitterness toward the child-protection system, as reflected in their stories.

Of the four parents in Group 2, Stephen and Jason had their contacts with their children supervised by their child-protection workers; Amelia had her contacts overseen by the child-protection agency, meaning that she was entitled to a maximum of six hours of unsupervised visits per month; and Carol was the only one allowed to keep her child overnight. The frequency of the two fathers' contacts with their children diminished during the months preceding the interviews, but for different reasons. In one father's case, the reason seems to be that he was in prison. In the other, it may have been that his contacts with his child were more difficult. But he said that he did not understand why child protection was constantly "cutting" his visitation rights. To sum up, the interviews with the parents in Group 2 indicate that their parenthood practice seems to be influenced by the framework and the flexibility associated with their contact arrangements. For example, Amelia said that being allowed only six hours of contact with her daughter limited the choice of activities that she could do with her, which was why they often just stayed home.

The parents in Group 2 also differed from the other parents as regards the themes related to parenthood exercise. In their interviews, Stephen and Amelia mentioned several times that they felt that the child-protection system did not hear them or take them into consideration. Amelia and Jason stressed that the child-protection workers responsible for their cases did not seem "to offered me workshops, I took them to show them that I was getting involved as a parent, and it was never enough". (Jason)

Carol told us that the child-protection agency had not helped her enough and had not guided her toward the resources available to her.

Lastly, unlike the other parents, the four parents in Group 2 referred to their biological connection to their children, perhaps out of a need they felt to assert their place, role, and legitimacy in their children's lives. For example, Jason said:

> And my little boy, I think that he's what I'm proudest of, he's my little clone. [He really looks] a lot like me. [...] I returned [with her], I had my little girl, my second little clone, girl version. My genes really won the battle there.

### 4.3. Group 3: Conjugal–Parental Dyad

Group 3, Conjugal–Parental Dyad, was more similar to Group 2 than to Group 1, particularly regarding the parent's relationship with the child-protection agency, although this relationship seemed more mixed in this case. What distinguished Group 3 from Groups 1 and 2 was a conjugal–parental relationship that raised other relationship issues. Group 3 comprised only one parent, Nicholas, who had two sons: one still living with his mother and the other placed with Nicholas's sister. Nicholas was still in a relationship with this second boy's mother and he was the only parent in our sample being in relation with the mother of the child placed. Moreover, in his story, he clearly shows how the relationship with the mother affects his relationship with his sister, who is the kinship caregiver for his son. He described his relationship with the kinship caregiver as mixed: he said that he had a good relationship with her, but also talked about conflicts that had arisen in the past and seemed to be continuing:

For sure, at the start I told her straight out, "He's not yours to have, he's ours." I told her, "I don't give a shit about you. I want my son, period." She didn't try to make him hers, but she did do whatever she could to get my son to call her "Mommy". (Nicholas)

He explained that he disagreed with his sister's having his son call her Mommy, because it seemed that by doing so, she was trying to take possession of him. Nicholas also described some disagreements that he had had with his sister about how to raise his son. Nicholas also reported that he sometimes felt torn between the demands made by his own family and those made by his in-laws (his child's mother's family). He explained that he had to act as a middleman, because his son had been placed with his own family rather than with his in-laws. This complex relationship dynamic, in which several persons were involved, differentiated this father from the parents in the two other groups.

Nicholas was also the only one of the nine parents interviewed who was still living with his children's other parent, so the continuation of this conjugal–parental relationship also distinguished him from the eight other parents in this study. Though he described his relationship with his conjugal partner as violent at times and marked by frequent, significant conflict, he also defended her, in particular against the child-protection agency, which had raised the possibility of their son's being placed for adoption. He said that this plan was completely illegitimate, because in his opinion, it would break the parental ties between his son and his partner permanently, though not those between his son and himself, because his son was going to be placed in his family. Nicholas saw his son twice per week, and his contacts were always supervised by the kinship caregiver or by Nicholas's mother (the child's paternal grandmother), who lived in the same building as the kinship caregiver.

Nicholas resembled the parents in the Parental Struggles group in his description of his fairly tense relationship with the child-protection agency. He talked at some length about his negative experience with this agency, and particularly its lack of recognition toward him.

## 5. Discussion

The results presented in this article reveal significant differences in the relationship dynamics experienced and reported by the nine parents interviewed. It must be stressed that none of the three groups into which we classified these parents was completely homogeneous. Although some strong tendencies could be discerned within each group, it was not unusual for one parent to differ from the others in the same group with respect to one or more of the themes. It should also be remembered that this typology is primarily exploratory. It attempts to identify some trends in the experience of parenthood in a very specific situation—when one's child has been placed in kinship care—and highlights the major impact that relationship issues have on this experience. Indeed, relationships are central to this typology, which is consistent with past studies that dealt specifically with kinship-care placements and regarded relationship dynamics as one of their distinctive features. It should also be noted that the present study is the first to analyze themes related to Houzel and Sellenet's three dimensions of parenthood and to draw connections between these dimensions and the relationship aspects of kinship-care placements.

The two major, contrasting patterns that this study found in relations between biological parents and kinship caregivers corroborate the findings of certain past studies, such as Strozier, Armstrong et al. [15], who also reported that this relationship was highly positive for some parents and highly negative for others. But all of the parents in that study were mothers whose children had been placed with their own mothers; in some cases, these authors report, the children's mothers did not like having them placed with their own mothers, whom they criticized in many ways. In the present study, all of the mothers whose children had been placed with their own mothers tended to report their relations with them as positive. Our study stands out by incorporating diverse filial bonds, not all of which involve a mother–daughter relationship between kinship caregivers and biological parents. In fact, our study notably reveals that mother–daughter bonds appear

more positive in contrast to filial bonds when the kinship caregiver is from the former in-law family. This finding highlights how placements within families can rekindle conflicts, particularly marital conflicts that resurface in the relationships between biological parents and kinship caregivers. In this regard, our study contributes to the existing knowledge on the subject and enhances the potential understanding of relationships within this type of placement.

The results of the present study also support the typology of kinship-care networks presented in that the relationship groups that we found in our analysis resemble three of the four types of care networks identified in O'Brien's [8]. First, in several respects, our Family Solidarity group resembles O'Brien's [8] "shared care networks": the parents co-operate well with the kinship caregiver and work together, without conflict, in the child's best interests, while the child-welfare agency stays on the periphery. But there are also some differences. In O'Brien's [8] shared care networks, the parents had voluntarily asked for their children to be placed with relatives, there was a history of informal care within the family, and the ages of the children at the time of placement and the time of the interviews varied widely. None of these was the case in any of the three groups identified in the present study. O'Brien [8] raises some interesting ideas about shared care networks, in particular the possibility that at an appropriate time, they might be released from the control of the child-welfare agency, whose role would then be limited to providing financial, social, and other appropriate forms of support and services.

Second, the relationships in our Parental Struggles group resemble those in O'Brien's [8] "distressed" networks. As she describes it, in their interviews, the parents in these networks tended to depict themselves as being excluded and, just like the parents in our Parental Struggles group, described experiencing conflicts both with the kinship caregivers and the child-welfare agency. O'Brien [8] points out that distressed networks are not usually present when a placement begins, because the child-welfare agency would not recommend a kinship-care placement under such circumstances. Instead, such networks arise as conflicts develop following the placement. This observation by O'Brien [8] makes a great deal of sense and is consistent with the findings in the present study, in which several parents reported that when their child was removed from the home, they had recommended the kinship caregiver with whom the child was placed, but that conflicts with this caregiver developed subsequently.

Third, the relationships experienced by the father who was the only member of our Conjugal–Parental Dyad group resembled those of the parents in O'Brien's [8] "oscillating networks", in which the conflicts oscillated between two patterns, according to whether the parent allied with the kinship caregiver or with the child-welfare agency. In our interviews with this father, a certain ambivalence emerged: sometimes he talked about allying himself with the kinship caregiver, but he also reported some major conflicts with her. O'Brien [8] states that when there is no explicit long-term care plan for their child, parents often find themselves in oscillating networks. Their ambivalence about the permanence of the placement and their hopes of getting their child back may cause their position to shift back and forth. For example, the father in the Conjugal–Parental Dyad group in the present study stated that he had felt very insecure when the child-protection agency had raised the possibility of a tutorship or adoption plan. This suggests that the alternative permanent care plan prescribed in the provincial framework on permanency planning [23] had not yet been made for his child.

The fourth type of kinship-care network described in O'Brien [8] is "quasi-adoption", and none of the parents in the present study seemed to fit into this category. This is not surprising, because the parents in this study had ongoing relationships with the child-protection agency, whereas in O'Brien's [8] study, this does not seem to have been the case. It would be interesting to observe how quasi-adoption networks may change over the long term. It should also be stressed that both in O'Brien [8] and in the present study, the findings represent "snapshots" of family situations as described and understood by the parents at a specific time in the course of the placement. It is quite possible that some

families may evolve from one type of network to another as the placement continues. In this regard, O'Brien [8] points out that at the beginning of placements, co-operative networks are more likely to be found.

The parents' relationships with the child-protection system also occupied a considerable place in the life stories of some of the parents whom we interviewed. Here too, the pattern seemed fairly polarized: some parents had a great deal to say about their experiences with the child-protection agency, the child-protection workers, and the services that they provided, while other parents had very little to say on these subjects. This finding leaves us with two potential hypotheses: either the youth protection services were not very involved in these situations, or the youth protection interventions were on the whole quite positive for the parents, so that they spoke very little about them. Also, those parents who reported a better relationship with the kinship caregiver had very little to say about the child-protection agency and seemed to perceive it as not very present or at least not very intrusive. In contrast, those parents who described their relationship with the kinship caregiver as more difficult had a lot to say about their difficult, negative experience with the agency's interventions.

The findings from this study add to current knowledge by describing certain details of the relationships in question from the parents' perspective. For example, if a child's parents separate, and the child is placed with his or her maternal grandmother, the father's relationship with this kinship caregiver may be affected by his separation from her daughter, and the father and the caregiver may play out some of the same conflicts that the couple had experienced before they separated. Ross, Cocks et al. [16] also support the idea that relationships between parents and kinship caregivers can be difficult, but state that child-protection agencies spend less time on managing such relationships than those between parents and regular foster families. In contrast, our analysis showed that when the relationship between the parent and the kinship caregiver was fairly negative, the child-protection agency actually seemed to have been more involved in managing their contacts and to have played a more prominent role in the parents' discourse. Similarly, the two types of conflictual care networks in the model of O'Brien [8] also illustrate the larger role that the child-protection system plays in such situations.

It is important to note certain limitations in this study. First, the sample was small, and its composition was affected by our having carried out the recruitment process in collaboration with the child-protection agency, so that parents who no longer had any ties with this agency could not be recruited. Though it is common for studies of this type to have small numbers of participants, the fact remains that no conclusions can be drawn about the saturation of the data. The life stories that the parents related in their interviews were highly varied and differed from one another, and the data do not suffice to achieve saturation of content on the various themes identified. It is essential to note, however, that saturation is not generally sought in this type of methodology. It is more common to conduct several interviews with each of the participants, in particular so as to reconstruct their stories as was done in the present study.

## 6. Conclusions

The findings presented here reveal some of the complexity of the family relationships that come into play when children are placed in kinship care, and the centrality of these relationships to this placement experience. Child-protection agencies should be alert to the various relationship issues underlying kinship-care placements and should try to provide the best possible support to the people concerned, including the children, who very often find themselves at the centre of the tensions that persist among the significant adults in their lives. This study has also shown that the parents of children placed in kinship care vary in their histories, their experiences, and their relationships with the other people involved in their children's placements, and that it is therefore important to adapt the clinical response to these parents' differing needs.

Clearly, kinship-care placements raise some interesting questions, in particular when the caregiver has been chosen for having a good relationship with the child but does not have a good relationship with the parent. What roles and responsibilities should the child-protection agency assume with specific regard to the various relationship issues that arise? The results show that, depending on the parents' perspective, they sometimes play a very active role, while in other situations they seem to be more on the sidelines. This raises questions about the relevance and necessity of youth protection services remaining active in situations similar to those of the family solidarity group, or whether alternative services might be in order.

Therefore, the results support the relational specificity of this type of placement and demonstrate the importance of offering interventions that take into account the different relational configurations, in order to support the various players throughout the placement and thus ensure not only the success of the placement, but above all the child's well-being.

Several aspects of kinship placement practice need to be clarified and improved, because these relationships play such a central role in kinship-care placements, and because it would be so difficult to tailor every intervention to the specific needs of every family concerned. A nuanced analysis of these needs is essential, but not sufficient. Much more must be done to make the services provided meet these needs—for example, provide a space where parents can tell their family histories, encourage meetings between kinship caregivers and parents to identify the relationship issues with which they are dealing, and support both of these parties by implementing effective communication strategies.

**Author Contributions:** The contribution of each author is as follows: the conceptualization of the research project was conceived by A.D. and validated by S.H. and M.-A.P., both supervisors. Interviews were conducted by A.D., as well as coding and analysis. Validation of the analysis was performed by M.-A.P. As for the writing of this article, it was carried out entirely by A.D., with S.H. and M.-A.P. reviewing, commenting, and revising the entire article. Integration of comments and revisions made by reviewers was conducted by A.D. and S.H. All authors have read and agreed to the published version of the manuscript.

**Funding:** This research was funded by the Fond de recherche du Québec-société et culture grant for doctoral student. The thesis project was also funded by grant from Institut Universitaire Jeunes en Difficulté and from School of social work, Université de Montréal.

**Institutional Review Board Statement:** The study was conducted in accordance with the Declaration of Helsinki, and approved by the Ethics Committee of the center intégré universitaire en santé et service sociaux du centre-sud-de-l'Île-de-Montréal for studies involving humans on 15 March 2018. # CÉR CJM-IU: 18-01-08.

**Informed Consent Statement:** Informed consent was obtained from all subjects involved in the study.

**Data Availability Statement:** For confidentiality reasons mandated by the research ethics committee, the data used for this article cannot be shared.

**Conflicts of Interest:** The authors declare no conflict of interest.

## Notes

[1]  This research project was approved by the CIUSSS-centre sud-de-l'île-de-Montréal research ethics committee. # of approbation 18-01-18.

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
