# Peer review of "An Exploratory Typology for Understanding Family-Relationship Issues in Kinship-Care Placements"

_societies, doi:10.3390/soc14030041_

Round 1

Reviewer 1 Report

Comments and Suggestions for Authors

This was an interesting and well written paper on an aspect of kinship placements that has not received extensive attention. Extant literature has tended to focus on child outcomes and either the significance and benefits of kinship placements or the difficulties from the kinship caregivers’ perspectives. This study’s more unique focus on the relational dynamics between biological parent and kinship caregiver was refreshing and extends work done by previous researchers as it adds more dimension to the relationships, including additional categories.

There are a few areas the author could attend to and/or offer clarification:

1. The assertion in the findings that the parents in group 1 had little involvement with child protection because not much was mentioned, seemed a bit of an overreach. Perhaps this relationship was not central to their life story?

2. It seemed interesting that parents in group 2 had their children placed with their former partner’s family and not their own. I wondered if their strained relationship would have been the same had it been their biological relative, as was the case with parents in group 1. Does the author have thoughts about this based on the interviews? Since parents in group 2 also had a negative relationship with child protection, could this have also influenced the dynamic with the kinship caregiver.

3. As I read the findings, I wondered about possible differences between groups 1 and 2 in the issues bringing their families into involvement with child protection and whether group 2 parents were regarded as perpetrators on abuse and /or neglect, as opposed to group 2 parents facing issues for which they were not necessarily deemed to be at fault. While Table 3 provides data that clarifies this, it would be useful to add some of this to the findings to help ground the perspectives shared.

4. I find the information on “filiation’ in Table 3 interesting. How relevant are these differences among the groups to the relational dynamic? Does the author see this as similar/related to O’Brien’s concept of feelings of exclusion by biological parents? Could this a contributing factor in the conflict?

Author Response

Thank you for your relevant comment and suggestion.

Reviewer 2 Report

Comments and Suggestions for Authors

Do you have the ages of the children? As mentioned in the document, you are missing a recruitment/sample paragraph and you may considering adding this element in. 

Author Response

(The authors gave the same response as above.)
